# Rapidly Progressive Idiopathic Pulmonary Arterial Hypertension in a Paediatric Patient Treated with Lung Transplantation

**DOI:** 10.3390/diagnostics13203185

**Published:** 2023-10-12

**Authors:** Filip Baszkowski, Weronika Pelczar-Płachta, Nikola Pempera, Sylwia Sławek-Szmyt, Marta Kałużna-Oleksy, Maciej Lesiak, Waldemar Bobkowski

**Affiliations:** 1Department of Pediatric Cardiology, Poznan University of Medical Sciences, 60-572 Poznan, Poland; 21st Department of Cardiology, Poznan University of Medical Sciences, 61-848 Poznan, Poland

**Keywords:** idiopathic pulmonary hypertension, bilateral lung transplantation, pulmonary vasodilators

## Abstract

Pulmonary arterial hypertension (PAH) is a rare heterogeneous disorder in the paediatric population which is mostly associated with congenital heart disease. The management of paediatric idiopathic PAH (IPAH) is difficult due to insufficient comparative data and depends on the results of evidence-based adult studies with several pulmonary vasodilators, as well as the clinical experiences of paediatric experts. Our aim was to present the case of a 9-year-old girl who underwent several methods of treatment, including pharmacotherapy with a significant reaction to treprostinil, as well as bilateral lung transplantation. The patient’s treatment was distinguished by the fact that the dose escalation was as rapid as that observed in the adult population. Due to the limited current evidence and knowledge, the initiation of treatment for these patients remains an individual choice. On the grounds of the number of non-specific symptoms, the diagnosis of this patient was a long process and based mainly on the differential diagnosis. The purpose of this paper is to study this example in order to highlight the importance of early symptoms and the quick implementation of intensive treatment. The applied methods may be useful in doubtful diagnosis processes and treatment procedures in the paediatric population.

A 9-year-old girl (30 kg, 138 cm) was admitted to the Cardiology Department with recurring heart palpitations and exertional dyspnoea. Upon admission, she was categorised in the II World Health Organization functional class (WHO-FC) and reached 413 m on the 6 min walk test (6MWT). Her N-terminal pro-B-type natriuretic peptide (NT-proBNP) serum concentration was increased to 1303 pg/mL (reference range: <391 pg/mL). An electrocardiogram (ECG) indicated sinus rhythm with a frequency of 110 beats per minute (bpm). Moreover, signs of right ventricle hypertrophy and overload were present in ECG.

Echocardiography (Figure 1) showed dilatation of the pulmonary artery ((PA) − 28 mm, z-score + 3.34), a diameter of the right ventricle in the upper limit of normal ((RV) diameter − 2.4 mm, z-score + 1.11), with high-velocity tricuspid regurgitation (4.3 m/s) and an elevated estimated RV systolic pressure ((RVSP) − 85 mmHg) and estimated mean pulmonary arterial pressure ((mPAP) − 59 mmHg). Right heart catheterisation confirmed precapillary pulmonary hypertension, with an mPAP of 49 mmHg, pulmonary artery wedge pressure (PAWP) of 12 mmHg, pulmonary vascular resistance of 12.2 Wood units (Wu), and a cardiac index of 3.45 L/min*m^2^, assessed by means of a thermodilution method. The acute vasoreactivity test with nitric oxide (dosage 20 ppm) was negative. The differential diagnosis to search for clear causes of pulmonary hypertension was performed according to the current guidelines [1]. The patient did not have any exposure to toxins or medications. AngioCT excluded a thromboembolic origin, and full rheumatological and infectious disease panels were performed, as well as molecular analysis. No pathological mutation was confirmed. A diagnosis of IPAH was made, and the patient qualified for combined PAH-targeted therapy. Full doses of bosentan (31.25 mg twice daily) and sildenafil (20 mg thrice daily) were administered, with no side effects. The patient’s condition stabilized for 3 months.

Four months after treatment initiation, the patient’s condition started to deteriorate. Moreover, the patient demonstrated central cyanosis associated with low arterial saturation (78%). A physical examination revealed bilateral, subscapular crackles and dyspnoea on minimal exertion (WHO-FC III/IV). Echocardiography showed the progression of the patient’s RV dysfunction with a further diameter enlargement to 3.3 cm (z-score + 2.36), dilatation of the PA to 3.1 cm (z-score + 4.3), and an increase in the assessed RVSP to 92 mmHg (Figure 2A). The quick progression of the disease after PAH-targeted therapy initiation, with the accompanying signs of pulmonary congestion, impaired gas exchange with a diffusing capacity of the lungs for carbon monoxide (DLCO) at 70%, and a suggestive CT image with centrilobular ground-glass opacities and lymphadenopathy, raised suspicion of pulmonary veno-occlusive disease (PVOD) as the underlying cause of the patient’s pulmonary hypertension [2]. After consultation with an expert transplant centre in Geneva (Hôpitaux universitaires de Genève), the PAH-specific treatment was suspended. The pulmonary vasodilators were withdrawn, and a diuretic treatment with furosemide (10 mg twice daily) and spironolactone (12.5 mg twice daily) was started. However, the patient’s condition continued to deteriorate. In the meantime, the patient had an additional consultation in Vienna (AKH University Hospital) with additional genetic tests, where no pathological mutation was found, and PVOD suspension was doubtful. A decision to reinitiate the PAH-targeted therapy with sildenafil (20 mg thrice daily) and bosentan (31.25 mg twice daily) was made. Finally, the patient was allocated to the waiting list for bilateral LTx (BLTx). Due to her continuing clinical deterioration, the patient qualified for triple therapy with a parenteral prostacyclin analogue. A continuous subcutaneous infusion of treprostinil was initiated and titrated at 2 ng/kg/min every 24 h at the dosage of 17 ng/kg/min. A more aggressive approach in dose escalation is supported by the current adult and paediatric data, but at the time, this option was not available in Poland [3,4,5,6]. However, Haarman et al. found that one year after upfront triple therapy had been started, a substantial number of children with IPAH were considered to need additional intervention, mainly due to their lack of improvement or clinical worsening [6]. After treprostinil administration, the patient’s clinical status improved (WHO-FC II, 6MWT distance 510 m), and this led to stabilization during the time spent waiting for BLTx, albeit without significant haemodynamic improvement. Only the RV diameter was found to be decreased in the echocardiographic assessment, with a drop to 2.7 cm (z-score + 1.44) 3 months after treprostinil administration (Figure 2B).

After the introduction of triple therapy, the patient underwent successful BLTx from a cytomegalovirus-positive donor in a referral transplant centre in Vienna. A cytological analysis of the postoperative material after bilateral lung transplantation finally excluded PVOD. Cardiac remodelling occurs early after BLTx as a result of a decreased RV afterload and elevated left ventricular volume preload due to the patient’s healthy lung vasculature and parenchyma of the graft [7]. The two-year follow-up was uneventful, with no signs of PH. In our patient, the echocardiography showed a progressive and rapid recovery of RV function without signs of tricuspid insufficiency (Figure 3B) and a decrease in the diameter (Figure 3A). The pulmonary artery was without valve insufficiency after transplantation, the mean flow volume (MFV) = 0.72 m/s, and we observed a complete withdrawal of PH features (Figure 3C).

The scans were carried with a GE Vivid E90 ultrasound machine before transplantation and with a Phillips Epiq CVx ultrasound machine at the two-year follow-up after transplantation, in accordance with the international recommendations [8].

## Figures and Tables

**Figure 1 diagnostics-13-03185-f001:**
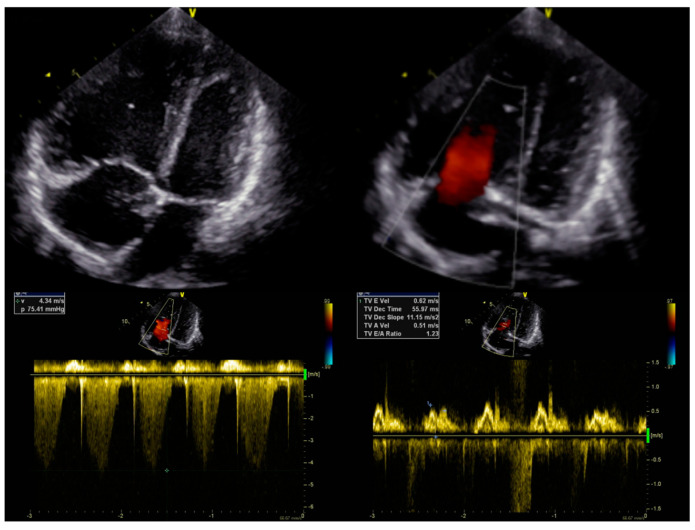
Initial echocardiography.

**Figure 2 diagnostics-13-03185-f002:**
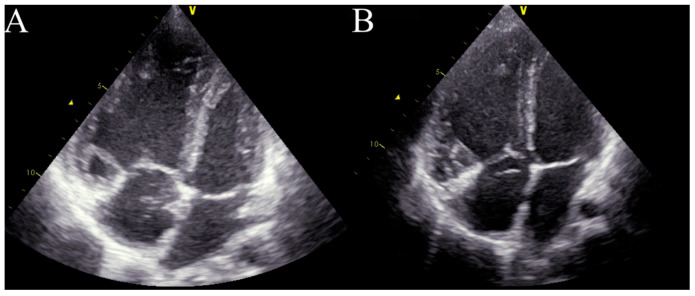
Echocardiography in critical care. (**A**)—right ventricle enlargement in the apical four-chamber view before treprostinil administration. (**B**)—right ventricle enlargement in the apical four-chamber view 3 months after treprostinil administration.

**Figure 3 diagnostics-13-03185-f003:**
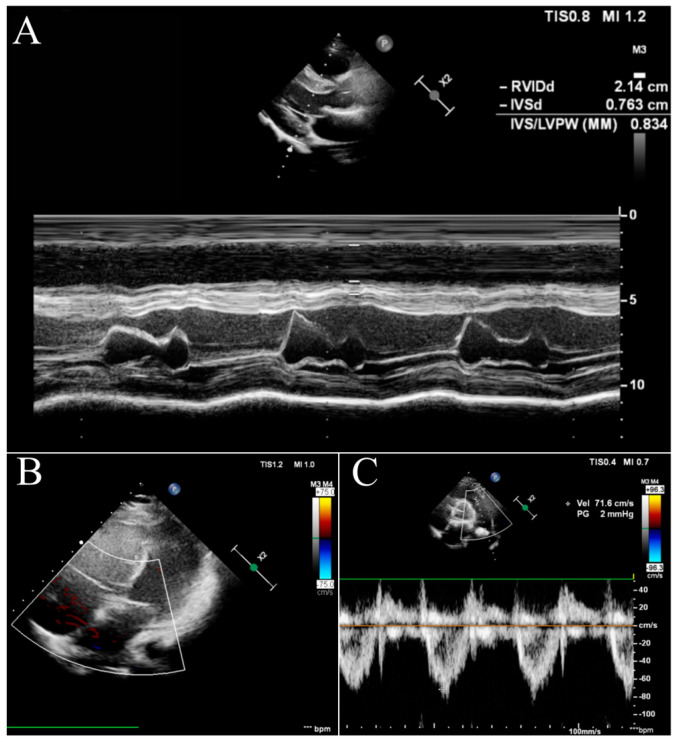
Echocardiography in the follow-up after BLTx. (**A**)—parasternal long-axis view in the M-mode. (**B**)—apical four-chamber view with doppler at the tricuspid valve. (**C**)—parasternal short-axis base view with doppler at the pulmonary valve.

## Data Availability

Data sharing not applicable.

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
