# Peer review of "Rapidly Progressive Idiopathic Pulmonary Arterial Hypertension in a Paediatric Patient Treated with Lung Transplantation"

_diagnostics, 2023, doi:10.3390/diagnostics13203185_

Round 1
Reviewer 1 Report
The authors have shown a clinical case of interest in the area of pediatric pneumatology with satisfactory surgical resolution. The reporting of unique cases is essential for the correct management of certain conditions and health disorders. However, notable deficiencies have been evident in the revised document.
Relatively poor quality of echocardiographic images. Furthermore, a lack of alignment is evident for the correct visualization and evaluation of the right cardiac chambers in the apical views.
It is recommended to expand the discussion on the final diagnosis of Idiopathic origin. No advanced imaging techniques, histological, cytological analysis, serology or molecular medicine were performed?
A higher level of comparison with previously published similar clinical cases is recommended.
It is recommended to provide an expanded profile of the patient (weight, height, body condition, etc.).
Lack of additional echocardiographic measurements to the hemodynamic assessment study, no measurements of systolic or diastolic functionality are reported. Alternative methods such as speckle tracking strain, tissue Doppler, etc. are not used.
Line 36: it is recommended to clarify the morphological term of hypertrophy. What were the signs of hypertrophy and overload? In which cardiac chambers or structures was it observed? Are there numerical values in this regard?
In general, the use of abbreviations should be reevaluated in the document. Sometimes they are used without previously being named (example line 46, IPAH, line 86: MFV) and on the other hand they are not used on various occasions (Example line 41; Precapillary pulponary hypertension).
It is recommended to review the echocardiographic views used with more details.
It is recommended that you specify geographic locations. Example line 63.
It would be appreciated to know the ultrasonographic equipment used.
Line 64: What was the diuretic therapy used? At what dose? There is an absence of details in the methodology used.
Based on what criteria was the presence of PVOD excluded?
Author Response
Dear Reviewer
Thank you for your revision. We have followed your suggestions and applied it in our work. I want to give you the answers, which you have submitted
Relatively poor quality of echocardiographic images. Furthermore, a lack of alignment is evident for the correct visualization and evaluation of the right cardiac chambers in the apical views.
Thank you for pointing this out. We have changed the 2B image to better alignment. What’s more, we have added additional figure with haemodynamic assessment of the RV showing decrease in diameters after transplantation.
It is recommended to expand the discussion on the final diagnosis of Idiopathic origin. No advanced imaging techniques, histological, cytological analysis, serology or molecular medicine were performed?
Idiopathic origin of the PH was made on the basis of the differential diagnosis. Our patient didn’t have any exposure to toxins or medications. AngioCT exclude thromboembolic origin, full rheumatological and infectious diseases panels were performed as well as molecular analysis. Each test was within normal result. This discussion can be found in lines 46-48.
A higher level of comparison with previously published similar clinical cases is recommended.
We are submitting our paper as “Interesting Image”. We tried to compare it with previously published similar examples sended us by editors.
It is recommended to provide an expanded profile of the patient (weight, height, body condition, etc.).
Agree. We have made an adjustment. Line 32
Lack of additional echocardiographic measurements to the hemodynamic assessment study, no measurements of systolic or diastolic functionality are reported. Alternative methods such as speckle tracking strain, tissue Doppler, etc. are not used.
We agree with this comment. We have added additional figure- 3A with decrease in RV diameter after transplantation (Parasternal short-axis view in M-Mode).
Line 36: it is recommended to clarify the morphological term of hypertrophy. What were the signs of hypertrophy and overload? In which cardiac chambers or structures was it observed? Are there numerical values in this regard?
Described signs of the right ventricle hypertrophy and overload were related to ECG alternations. We have adjusted this in our paper.
In general, the use of abbreviations should be reevaluated in the document. Sometimes they are used without previously being named (example line 46, IPAH, line 86: MFV) and on the other hand they are not used on various occasions (Example line 41; Precapillary pulponary hypertension).
Thank you for pointing this out. We have made and adjustments in line 93: Mean flow volume (MFV) . However, abbreviation IPAH was used in abstract (Line 14). That is the reason, why we are using it then without explanation. We also decided to not use abbreviations for precapillary pulmonary hypertension. We are using it only once in our paper and the abbreviation is not commonly used in literature.
It is recommended to review the echocardiographic views used with more details.
We decided to limit the review of the echocardiographic views to the alternations. Images had been done during therapeutic process, which lasted 4 years. Addition of the parameters where result was within normal range could be illegible.
It is recommended that you specify geographic locations. Example line 63.
Agree. We have, accordingly, added name of the hospitals in line 65 and 68.
It would be appreciated to know the ultrasonographic equipment used.
Thank you for pointing this out. We have, accordingly, added Lines 95-97 and reference 8.
Line 64: What was the diuretic therapy used? At what dose? There is an absence of details in the methodology used.
Agree. We have added "furosemide (10 mg twice daily) and spironolactone (12,5 mg twice daily)" accordingly in line 66-67
Based on what criteria was the presence of PVOD excluded?
Thank you for pointing this out. PVOD was firstly doubted after patient visit in Vienna. Radiological scan was not clear for the experts in Vienna as well as DLCO result, which was only slightly reduced (70%). In the clinical status of the girl PVOD diagnosis should follow severe reduction in the diffusing capacity with DLCO <50%. Additional genetic tests also exclude presence of EIF2AK4 gene for hereditary PVOD. On the basis of these investigation and quick deterioration of clinical status Treprostinil treatment was initiated. To be sure about PVOD, exclusion biopsy of the lung would had been needed. However, it is contraindicated by the current guidelines and was impossible in the current stage of the patient. The cytological analysis of the postoperative material after bilateral lung transplantation give the final answer and excluded PVOD. However, we are not able to receive this images due to privacy law differences between countries.
We have added this information in lines 87-88. We also modified discussion on that issue in lines 68-69.
New, corrected version of manuscript can be found in attachment. Changes are highlight in red.

Reviewer 2 Report
1. In the text, it is said that IPAH differentials were ruled out. However, despite the treatment, it did not improve. It would be worth knowing if TEP was ruled out with pulmonary angioTAC because HP treatment differs in that context (Riociguat, for example).
2. It is said that the NT-proBNP was found and that “clinically” rales were found. Was a pulmonary ultrasound done to ensure that it was pulmonary congestion? Was Vexus performed to value systemic congestion?
3. most of the time, the rales are associated with left ventricular dysfunction. How was the left ventricular ejection fraction (LVEF)?
4. The last image seems like a parasternal left short axis at the level of large vessels. It serves to estimate lung pressure. However, it would be worth seeing if the dilation of the VD was reduced compared to the previous images with a view of 4 cameras (I think it is the A, but it is not a good image).
Author Response
Dear Editor
Thank you for your revision. We have followed your suggestions and applied it in our work. I want to give you the answers, which you have submitted:
- Thank you for pointing this out. Pulmonary thromboembolism was, in fact, ruled out. Pulmonary CT angiography showed no signs of any the thromboembolic event. However, CT showed centrilobular ground-glass opacities and lymphadenopathy, what pointed us towards the presence of PVOD/PCH. These suspiction accompanied whole diagnostic and therapeutic process. Nevertheless, in Vienna PVOD diagnosis was doubted. It could be explanation for poor response to sildenafil + bosentan treatment, but as well as some of the IPAH forms. Moreover, DLCO results (70%) didn’t accompany with the patient clinical status. PVOD diagnosis should have followed severe reduction in the diffusing capacity with DLCO <50%. Current guidelines does not recommend lung biopsy in the diagnostic process. In fact, it was not fully clear until cytological analysis of postoperative material after bilateral lung transplant whether some type of IPAH or PVOD is the final diagnosis. Treprostinil were, in fact, administered empirically after multicenter counsultations. Quick deterioration, long queue to BLTx and lack of options forced us to attempt treprostinil treatment, which was successful.
We have clarified it in our manuscript (Line 46 and 47). Information about CT scan was also added. (Line 62-63) - Ultrasonography of lungs showed modest amount of fluid, which disappeared after furosemide and spironolactone (2x 10mg, 2x 12,5mg respectively) administration as well as other congestion features (rales, cyanosis, dyspnoea and improvement in saturation from 78% to 92% during oxygen therapy). No sign of systemic congestion was spotted (IVC < 2cm).
Information about exact diuretic treatment was added (Line 66). - Left ventricular ejection fraction were all of the time within normal range (LVEF=78%).
- Unluckily, there is no proper apical 4CH saved in hospital’s archive. However, we have added additional figure with visualisation of the right ventricle (PSAX in M-Mode) to better to show decrease in the RV diameter(Fig.3A). Figure 3B (4CH with doppler) was added to show lack of tricuspid insufficiency. (Fig. 3; Lines 90-93)
Please see the attachement with the new version of the manuscript. Changes were highlighted with red.

Round 2
Reviewer 1 Report
Most of the recommendations have been evaluated and the document has improved substantially.